# A universal approach for the synthesis of two-dimensional binary compounds

Abhay Shivayogimath [1,2], Joachim Dahl Thomsen[1,2], David M.A. Mackenzie [2,3], Mathias Geisler[2,4], Raluca-Maria Stan[5], Ann Julie Holt[5], Marco Bianchi[5], Andrea Crovetto[6], Patrick R. Whelan [1,2], Alexandra Carvalho[7], Antonio H. Castro Neto[7], Philip Hofmann[5], Nicolas Stenger [2,4], Peter Bøggild [1,2] & Timothy J. Booth [1,2]

Only a few of the vast range of potential two-dimensional materials (2D) have been isolated or synthesised to date. Typically, 2D materials are discovered by mechanically exfoliating naturally occurring bulk crystals to produce atomically thin layers, after which a material-specific vapour synthesis method must be developed to grow interesting candidates in a scalable manner. Here we show a general approach for synthesising thin layers of two-dimensional binary compounds. We apply the method to obtain high quality, epitaxial $MoS_2$ films, and extend the principle to the synthesis of a wide range of other materials—both well-known and never-before isolated—including transition metal sulphides, selenides, tellurides, and nitrides. This approach greatly simplifies the synthesis of currently known materials, and provides a general framework for synthesising both predicted and unexpected new 2D compounds.

[1] DTU Physics, Technical University of Denmark, Ørsteds Plads 345, DK-2800 Kgs Lyngby, Denmark. [2] Centre for Nanostructured Graphene (CNG), Technical University of Denmark, Ørsteds Plads 345C, DK-2800 Kgs Lyngby, Denmark. [3] Department of Electronics and Nanoengineering, Aalto University, P.O. Box 13500, FI-00076 Aalto, Finland. [4] DTU Fotonik, Technical University of Denmark, Ørsteds Plads 343, DK-2800 Kgs Lyngby, Denmark. [5] Department of Physics and Astronomy, Interdisciplinary Nanoscience Center (iNANO), Aarhus University, 8000 Aarhus, Denmark. [6] DTU Physics, Technical University of Denmark, Fysikvej 312, DK-2800 Kgs Lyngby, Denmark. [7] Centre for Advanced 2D Materials and Graphene Research Centre, National University of Singapore, 2 Science Drive 3, Singapore 117542, Singapore. Correspondence and requests for materials should be addressed to T.J.B. (email: tibo@dtu.dk)

Many of the possible 2D materials are binary compounds of the form $MX_n$, where $M$ is typically a transition metal and $X$ a chalcogen or non-metal from groups IV, V, or VI[1–5]. The molybdenum and tungsten disulphides and diselenides remain the most commonly studied 2D binary compounds—other than hexagonal boron nitride (hBN)—due to the ready availability of naturally occurring bulk crystals amenable to exfoliation. Chemical vapour deposition (CVD) techniques for the scalable synthesis of these materials are available[6]; however, controlling the stoichiometry and hence the defect density can be challenging. Such techniques typically employ solid metal oxide[7–10] or metal-organic[11] precursors which are chalcogenated at elevated temperatures. Finding appropriate metal precursors can be a limiting challenge for extending these methods to other 2D transition metal compounds, and as a result requires single-process dedicated equipment that is highly optimised for growing one specific material. A more general method using simpler precursors would thus be beneficial. Published CVD growth models for binary compounds on metal catalysts stipulate that both $M$ and $X$ elements be insoluble in the catalyst to ensure surface-limited growth[12–14], by analogy with CVD graphene growth on copper[15]. In fact, the synthesis of monolayer hBN films on copper—despite the high solubility of boron in copper[16]—suggests that only one component need be insoluble.

Here we present a general method for synthesising two-dimensional compounds on a metal catalyst from solid elemental precursors (Fig. 1). We arrange a single component solid solution, as used for hBN growth on copper, by alloying metal $M$ films with gold, which has limited solubility of the $X$ elements ($X$ = S, Se, Te, N). In brief, a thin layer (~20 nm) of metal $M$ is sputtered onto a $c$-plane sapphire substrate followed by a thick layer (~ 500 nm) of gold (see Methods). The $M$-Au layer is then heated to 850 °C to form an alloy with an Au {111} surface. The relative thicknesses of the $M$ and Au layers determines the concentration of $M$ in the final alloy, which here is deliberately limited to ≤5 at%[17–24] in order to maintain single-phase alloying conditions. The Au-$M$ alloy is subsequently exposed to a vapour-phase precursor of element $X$. The limited solubility of $X$ in the gold restricts the formation of $MX_n$ compounds to the surface of the alloy—at the solid–gas interface—resulting in few-atom thick layers of binary compounds that are epitaxially aligned to the underlying Au substrate.

The process enables the epitaxial synthesis of both known and new 2D materials using a single recipe and simple elemental precursors, demonstrated here by the synthesis of 20 compounds including sulphides, selenides, tellurides, and nitrides.

## Results

**Synthesis and characterisation of MoS₂.** To benchmark this approach we first synthesise and characterise $MoS_2$ layers (Fig. 2). Individual domains display triangular morphology as seen from scanning electron microscopy (SEM) images of the catalyst surface after growth (Fig. 2a). The gold catalyst adopts a {111} surface on the <001> sapphire substrate after annealing (Supplementary Fig. 1), leading to epitaxial growth of $MoS_2$ across the catalyst surface. Low-energy electron diffraction (LEED) (Fig. 2c) and angle-resolved photoemission spectroscopy (ARPES) (Fig. 2d) of as-grown $MoS_2$ domains on gold confirm uniform epitaxial alignment across the underlying substrate, as evidenced by the moiré satellite peaks visible in LEED[25] and well resolved bands in ARPES. ARPES also indicates that the $MoS_2$ domains are primarily monolayers, based on the absence of a strong anti-bonding band at the $\Gamma$ point that is characteristic of multilayer formation[26,27]. Raman spectroscopy with 455 nm excitation of $MoS_2$ crystals transferred to an

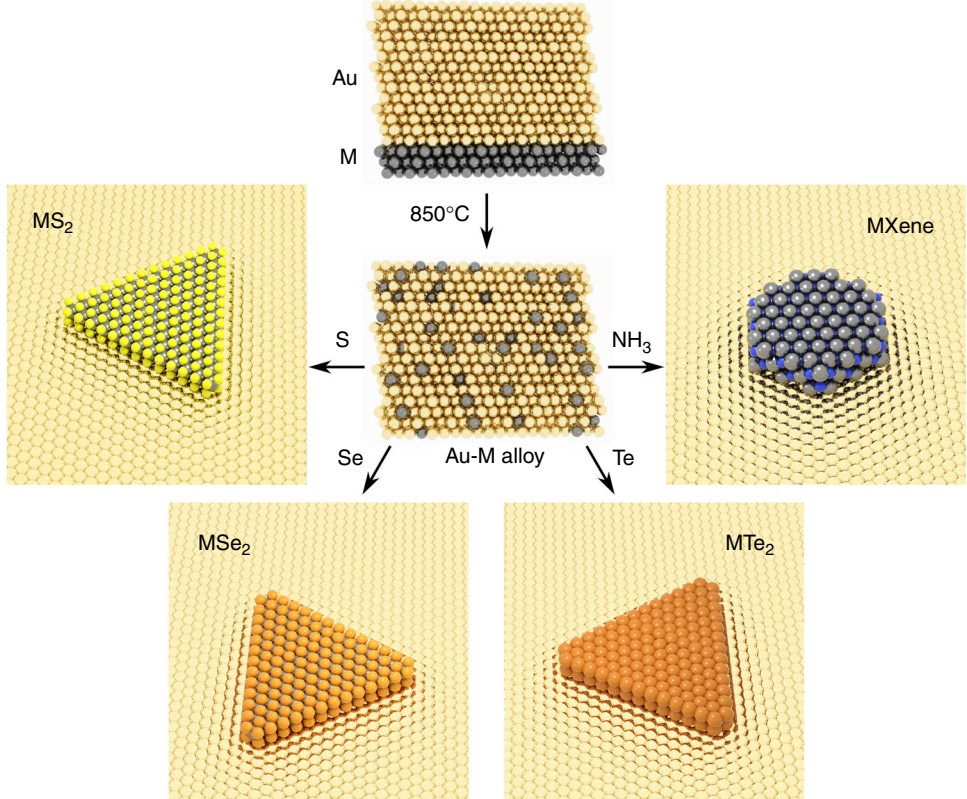

**Fig. 1** Schematic overview of the synthesis process. A thin layer (~20 nm) of metal $M$ is sputtered onto a $c$-plane sapphire substrate, and a thick layer (~500 nm) of Au is sputtered on top. The sample is annealed at 850 °C to produce an Au-$M$ alloy, which is then exposed to a vapour of S, Se, Te, or more generally an elemental $X$ gas or vapour. The growth of binary $MX_n$ compounds proceeds at the surface of the Au-$M$ layer and is surface-limited

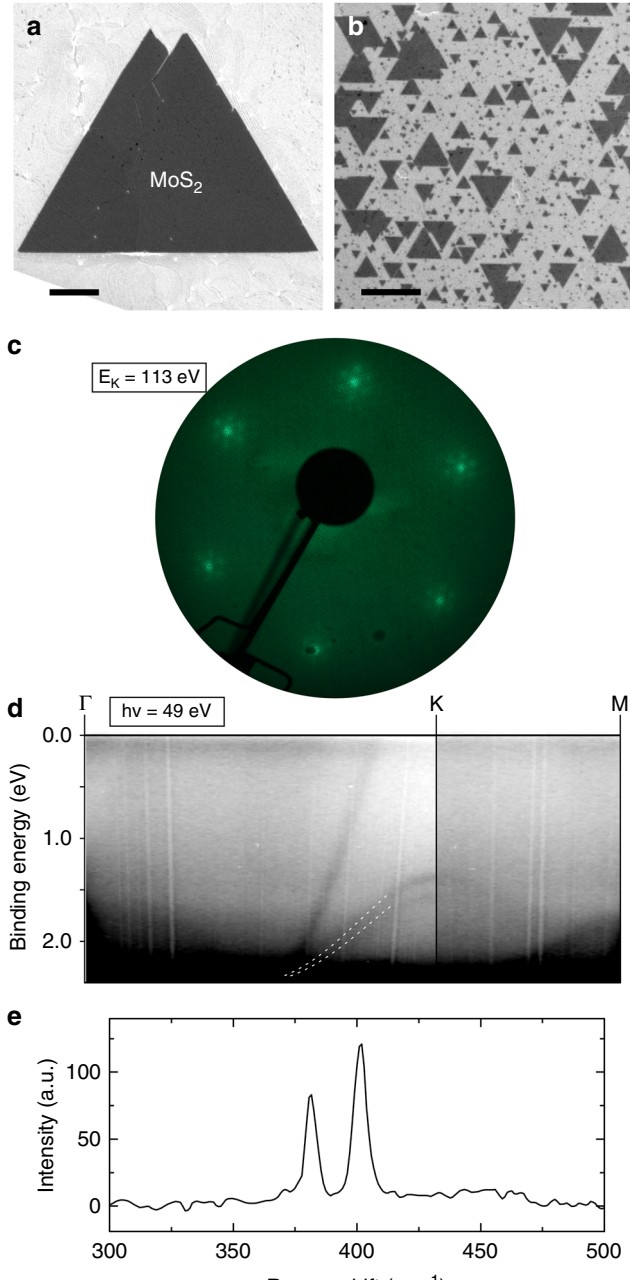

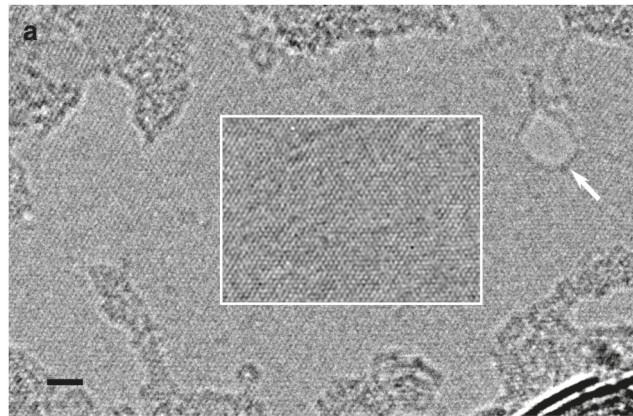

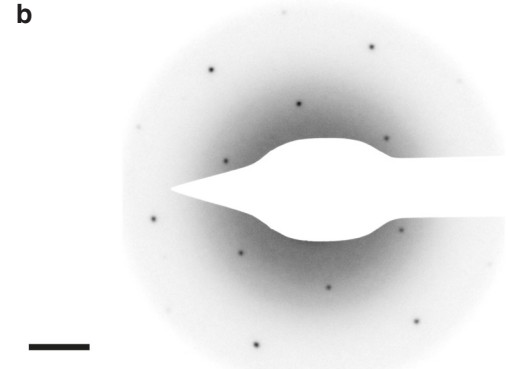

**Fig. 2** Characterisation of $MoS_2$. **a** SEM image of an individual $MoS_2$ domain on Au {111} catalyst. Scale bar 1 μm. **b** SEM of epitaxially oriented $MoS_2$ domains on gold. Scale bar 1 μm. **c** LEED pattern and **d** ARPES of as-grown $MoS_2$ on Au {111}. The dashed lines in **d** serve as visual markers of the valence band of monolayer $MoS_2$ at the K point of the Brillouin zone. **e** Raman response with 455 nm excitation of $MoS_2$ transferred onto 90 nm $SiO_2$/Si substrate. Two peaks are evident, the $E_{2g}$ peak at 381.40 $cm^{-1}$ and the $A_{1g}$ at 401.40 $cm^{-1}$

**Fig. 3** TEM characterisation of $MoS_2$. **a** High-resolution TEM of suspended $MoS_2$. A hole in the monolayer introduced by knock-on damage during imaging is indicated (white arrow), where lattice fringes are absent. The boxed region has had an iterative nonlinear denoising filter applied to highlight the $MoS_2$ lattice and reduce shot noise[50]. Scale bar 2 nm. **b** SAED pattern of suspended $MoS_2$ layers with a 100 nm diameter aperture showing single crystal long range order. Scale bar 1 $nm^{-1}$

oxidised silicon substrate shows two peaks, the $E_{2g}$ and $A_{1g}$ at 381.40 ± 0.05 and 401.40 ± 0.04 $cm^{-1}$ respectively (Fig. 2e). While the positions and intensities of these peaks can in general vary as a result of strain and doping, their separation of ≈ 20 $cm^{-1}$ is diagnostic of monolayer $MoS_2$[28].

High resolution transmission electron microscopy (TEM) and selected area electron diffraction (SAED) images of crystals transferred to holey carbon support grids confirms the crystal structure of the $MoS_2$ layers (Fig. 3a, b) and that the layers are free of atomic defects over the areas imaged. A region where vacuum is visible through the sample due to knock-on damage is indicated in Fig. 3a, confirming that the suspended region is a monolayer.

Figure 4a shows photoluminescence (PL) measurements (see Methods) of $MoS_2$ domains transferred onto 90 nm $SiO_2$ on Si substrates (solid line), compared to a monolayer exfoliated $MoS_2$ crystal control sample (dashed line). Identical acquisition parameters and substrates were used in both cases. The intensity of the PL response is comparable in both cases, while the PL peak for transferred crystals is blue-shifted with respect to exfoliated samples. Electric field-effect measurements were performed on 100 μm × 100 μm unencapsulated devices of continuous $MoS_2$ films transferred onto 300 nm $SiO_2$ on Si substrates with pre-patterned contacts (see Methods). Four-point sheet conductivity $σ_S$ was calculated at varying gate bias ($V_G$) as described in ref. [29], with the results for a representative device shown in Fig. 4b. The device shows an on-off ratio of >$10^4$ over the gate bias range, and small maximum hysteresis of about 16 V. The low level of hysteresis suggests that the overall number of charge traps is low, including intrinsic charge traps within $MoS_2$ itself[30]. The field-effect mobility $μ$ was calculated using the formula[31] $μ = (dσ_S/dV_G) \cdot 1/C_{ox}$, where $C_{ox}$ is the capacitance per unit area of the back gate. Our measured devices showed a range of $μ$ between 5 and 30 $cm^2 V^{-1} s^{-1}$ (Supplementary Table 1), which is comparable to results for unencapsulated exfoliated $MoS_2$[31].

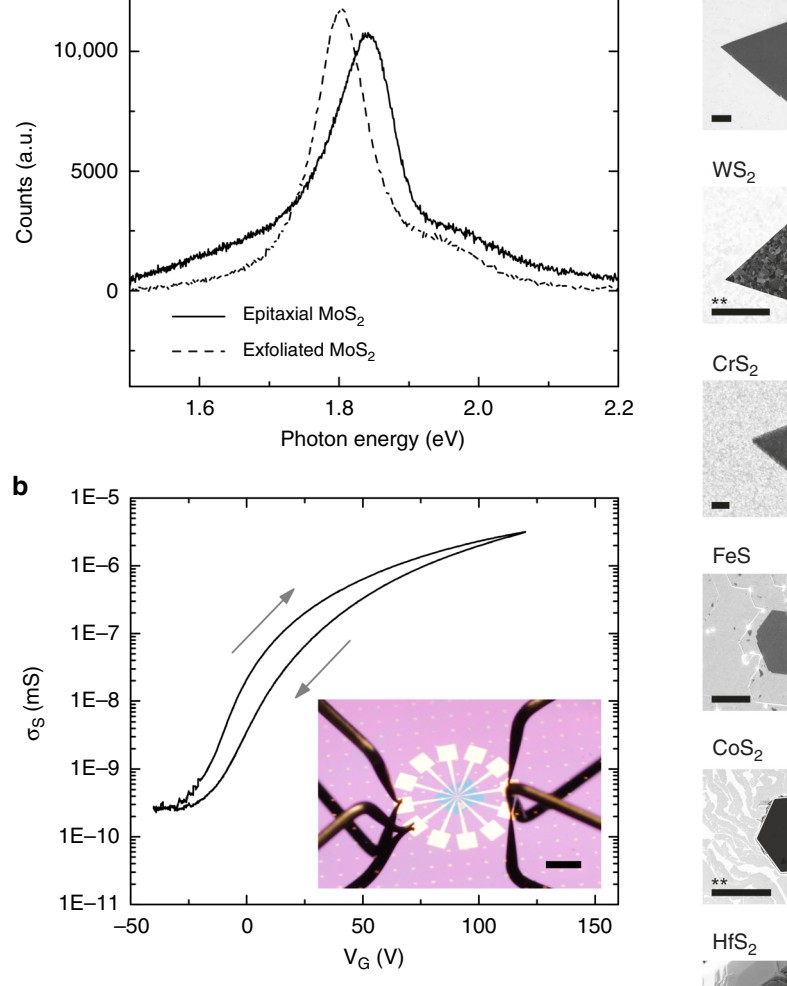

**Fig. 4** Photoluminescence and electrical characterisation of MoS₂.
**a** Photoluminescence spectra for the as-grown MoS₂ transferred to SiO₂ (solid line) vs. mechanically exfoliated monolayer MoS₂ (dashed line).
**b** Gate-dependent conductance measurements of a representative unencapsulated MoS₂ device. Inset: image of a measured device. Scale bar 1 mm

**Synthesis of other binary compounds**. Varying $M$ or $X$ for a range of different elements ($M$ = Mo, W, Cr, Fe, Co, Hf, Nb, V; $X$ = S, Se, Te) under identical growth conditions results in the structures visible in the SEM images in Fig. 5. X-ray photoelectron spectroscopy (XPS) data confirming the expected stoichiometry and bonding for each of these binary transition metal dichalcogenides is presented in Supplementary Figs. 2–18, where further characterisation on selected materials is also presented. We have also shown the growth of select transitional metal nitrides, presented in Supplementary Figs. 19–21, along with calculated band structures (Supplementary Fig. 22) for selected novel materials not present in the literature.

## Discussion

We have shown the synthesis of epitaxially aligned MoS₂ layers whose properties—namely Raman spectroscopic response, nanoscale crystalline structure, intensity of PL response, and electric field-effect properties—are comparable to mechanically exfoliated monolayers from bulk crystals or high-quality films from more complex CVD processes. This process is selective towards monolayer synthesis despite the fact that the solubility of

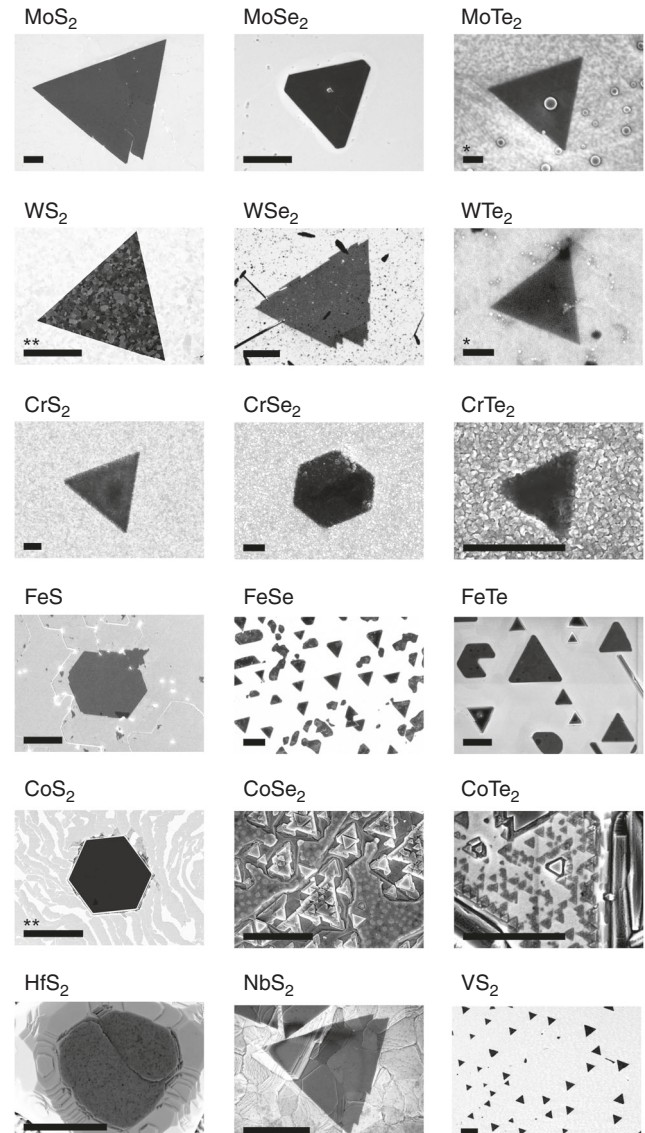

**Fig. 5** Library of layered transition metal chalcogenides. SEM images of the various transition metal chalcogenides grown by the present method. All presented materials are grown under identical process conditions, varying only $M$ and $X$. Further characterisation for the materials is presented in Supplementary Figs. 2–18. Scale bars are 1 μm except where marked: *: 100 nm, **: 10 μm

molybdenum in gold is appreciable (~1 at%) at 850 °C[17], demonstrating that the strict insolubility of all component elements is not required when extending surface-limited growth models to multi-elemental 2D compounds.

Furthermore, we show XPS and SEM data for a wide range of other materials produced simply by selecting different pairs of $M$ and $X$ elemental precursors, and additional Raman, atomic force microscopy (AFM), PL, and TEM characterisation for the materials which are stable during transfer to oxidised silicon or on the catalyst layer. Without further optimisation, a large selection of materials which display morphologies, stoichiometries, and chemical bonding expected of binary 2D transition metal dichalcogenides can be grown. While exhaustive characterisation of all of these materials is beyond the scope of this manuscript—in particular for candidate materials which have not previously been synthesised or isolated from bulk crystallites before—the simplicity of this growth strategy enables very rapid

experimentation in terms of the range of precursor combinations that can be tested and materials which can be grown. In practice since multiple $M$-Au alloy samples can be simultaneously exposed to the same $X$ precursor, the time needed to fully characterise such materials individually far exceeds the time needed to produce them in parallel.

The surface-limited growth of a wide variety of binary 2D materials presented in this work relies on the use of gold as a catalyst layer. Gold readily alloys with most transition metals $M$, but shows limited solubility for $X$ elements at our growth temperatures. Moreover, gold is unique in that it does not react with $X$ precursors at these temperatures. Gold is catalytically active, which aids in the formation of crystalline atomically thin materials, and the {111} surface formed on $c$-plane sapphire also templates the epitaxial alignment of the grown materials. The method presented here is distinct from vapour–liquid–solid (VLS) and vapour–solid–solid (VSS) growth models[32] of III–V nanowires since we actively avoid supersaturation of component metals in gold—a key hallmark and necessary prerequisite for VLS/VSS growth mechanisms—by limiting the amount of metal available to <5 at%. This ensures that the resulting alloy behaves as a single-phase, simple solid solution at the growth temperature. The role of the gold here is to facilitate phase separation of $M$ and $X$ precursors such that they react only at the vapour–solid interface. Once this interface has been passivated by the 2D $MX_n$ layer, the growth terminates, similar to the case of graphene on copper.

The use of gold as a catalyst for synthesising 2D TMD layers has been previously reported in literature[12,13,33–35]; however, a key advantage of the present scheme is that solid elemental metal films are used as precursors rather than metal-organic vapours or volatile metal compounds, which makes our process significantly simpler to implement for both known and new TMDs. We also note that similar methods for growing atomically thin carbides and sulphides have been previously reported[36,37], but only for systems where both $M$ and $X$ components have limited solubility (<0.1 at%) in the catalyst. These represent a subset of the general approach presented here, i.e. one where the solubility of $M$ in the catalyst approaches zero. In such cases, growth is dominated by surface-mediated diffusion of both $M$ and $X$. Here instead the growth is dominated by precipitation of component $M$ from the catalyst bulk upon reacting with $X$ at the solid–gas interface at the surface of the catalyst. The crucial point is that surface-limited growth does not require the metal precursor to be insoluble in the catalyst layer. This fact allows our approach to be compatible with all precursor metals, in contrast to previous reports.

In the presented work, growth targeted at complete $MoS_2$ monolayer coverage—achieved here by increasing the growth temperature—can result in adlayers. In practice, adlayer growth can be reduced by tuning the growth temperature and time, reducing the flux of $X$, or limiting the alloy $M$ content of the gold catalyst layer. Controlling the $M$ content can be particularly important in determining the layer coverage, maintaining single-phase alloying conditions, and in preventing precipitation of excess metal upon cooling. Such precipitation can interfere with 2D growth and lead to 3D structures on the surface, as seen for Co[21] or Fe[20]. In general we expect that optimisation of additional deposition and growth parameters for individual $M$–$X$ combinations will be necessary, as has been the case for growth strategies for other 2D materials.

The present approach holds a number of advantages over state-of-the-art salt-assisted[38] or metal-organic chemical vapour deposition (MOCVD) of 2D materials[11]. Notably, solid elemental precursors (elemental metal thin films; vapour phase S, Se, or Te) or simple compounds ($NH_3$) are the only feedstocks required to grow the presented materials, which are in general more readily available and easier and safer to handle than metal-organic precursors and lead to less contaminants (such as carbon) being incorporated into the films. Critically, this approach provides a universal method for epitaxial synthesis of 2D materials without the need for salt additives[38], which may have a detrimental impact on material performance through loss of epitaxy and alkali metal doping[39]. While we have employed a tube furnace operating under low pressure, the growth scheme presented does not rely on a flow but simply the presence of $X$ precursor and as such can be performed in a sealed chamber. Such a scheme might be particularly beneficial to reduce the amount of oxygen and water in the growth system—the alloys of certain transition metals (e.g. Cr, Ta, Nb, Hf, V) are highly sensitive to oxidising impurities in the growth chamber[40,41], which can interfere with growth by passivating the catalyst surface with metal oxides. This issue can be addressed by operating in a highly reducing environment or under high vacuum.

Some of the materials grown are unstable under ambient conditions—no special steps were taken to limit the exposure of samples to ambient atmosphere before characterisation. As such, transfer of many of these materials to, e.g., oxidised silicon substrates or TEM grids is challenging, as they rapidly degrade on contact with aqueous or oxygen-containing solutions, which has hampered extended characterisation in many cases. Recent progress in the solution-phase exfoliation of air and water-sensitive 2D crystals[42–44] suggests that similar strategies might be successfully employed here.

## Conclusion

In summary, we have demonstrated a simple and universal strategy for approaching the growth of few-atom thick binary compounds, including transition metal mono- and dichalcogenides and nitride $MX$enes, based on the insight that only one component of a binary compound need be insoluble to achieve surface-limited growth. The strategy employs only a volatile $X$ precursor, while all the presented materials are grown under identical conditions. Notably, this method enables the growth of epitaxially oriented $MX_n$ layers on Au. New 2D materials can be made through the free choice of $M$ and $X$, and different compounds can be obtained on the same growth substrate simply by switching the precursor $X$ gas. As such, this scalable growth technique simplifies the production of existing binary 2D materials with a quality comparable to exfoliated crystals, and at the same time greatly increases the range of such materials available. We do not doubt that growth conditions for materials can be individually optimised, and that with further research, growth of in-plane and out-of-plane heterostructures could also be accessible.

## Methods

**Preparation of gold-metal $M$ substrates.** Substrates were prepared by physical vapour deposition of a thin layer (~20 nm) of metal $M$ followed by a thick layer (between 300 nm–1 μm, typically ~500 nm) of gold (Lesker, 99,999%) on a <001> sapphire substrate. Oxidation of the metal $M$ is avoided by immediate encapsulation with gold before exposure to ambient conditions.

**Synthesis of 2D transition metal chalcogenides and nitrides.** Transition metal chalcogenides were synthesised in a hot-wall quartz tube reactor under low-pressure conditions. The chamber was flushed three times with argon (Ar), and the samples were subsequently heated to 850 °C under 100 sccm Ar. The samples were annealed at this temperature for 30–60 min, and growth was subsequently carried out for 10–15 min by exposing the samples to volatised chalcogen $X$ vapours. The vapours were generated by heating solid chalcogen precursors situated upstream from the samples: ~110 °C for sulphur (sulphur flakes; Sigma Aldrich), ~220 °C for selenium (selenium pellets; Sigma Aldrich), and ~420 °C for tellurium (tellurium pieces; Sigma Aldrich). After growth, the samples were naturally cooled to room temperature under 100 sccm Ar flow. Synthesis of continuous $MoS_2$ films for

electrical device measurements was achieved by increasing the growth temperature to 950 °C.

Identical processing conditions were used to synthesise transition metal nitrides in a cold-wall reactor (AIXTRON Black Magic), except that the entire process was done under 100 sccm $H_2$ flow instead of argon in order to mitigate surface oxidation of the alloys. The growth of nitrides was performed by introducing 5 sccm $NH_3$ into the chamber for 5 min The samples were then naturally cooled down to room temperature under 100 sccm $H_2$.

**Transfer of 2D materials**. Samples were transferred from the gold substrate by etching transfer. A solution of 10% wt. PMMA 950K in anisole (Sigma Aldrich) was spincoated onto the samples at 1500 rpm for 1 min, after which the samples were baked at 160 °C for 15 min. The polymer film was then manually removed at the edges of the sample. The samples were then put in a $KI/I_2$ gold etchant solution (standard gold etchant; Sigma Aldrich). After the gold was completely etched, the films were washed in DI water and transferred onto oxidised silicon substrates, where they were baked at 160 °C for 10 min. PMMA was subsequently removed in acetone. Transfer of 2D materials onto TEM grids (Quantifoil GmbH) was done by wedging transfer[45,46] from transferred films on oxidised silicon.

**Crystal and band structure characterisation**. SEM images were taken in a Zeiss Supra 40VP operated in in-lens detection mode at 5 keV. TEM characterisation of transferred $MoS_2$ was done in an FEI Tecnai T20 G2 operated at 200 kV. ARPES and LEED measurements were conducted at the SGM-3 beamline endstation at ASTRID2 in Aarhus, Denmark. ARPES measurements for $MoS_2$ were acquired at $T = 30$ K and $hv = 49$ eV, using an energy resolution <25 meV and angular resolution <0.2°[47]. LEED images were acquired at $T = 30$ K and $E_K = 113$ eV.

**Raman and PL measurements**. Raman spectroscopy was conducted in a Thermo Fisher DXR microscope equipped with a 455 nm laser. Measurements were made using an incident power of 5 mW and a 50× objective, and 5 acquisitions with 10 s exposure time were collected for each Raman spectrum. Photoluminescence spectra were obtained using a custom spectroscopy setup built from a Nikon Eclipse Ti-U inverted microscope. The excitation source was a 407 nm diode laser from Integrated Optics. The light was focused to a diffraction limited spot on the sample with a TU Plan Fluor objective from Nikon (×100, 0.9 NA) resulting in an incident power of 30 μW. The emitted fluorescent light was collected with the same objective, and the spectra were recorded using a Shamrock 303i Spectrometer equipped with a 450 nm longpass filter (FELH0450 from Thorlabs) and an electronically cooled Newton 970 EMCCD. A total of five acquisitions with 1 s exposure time each were collected for each PL spectrum.

**Fabrication of electrical devices**. Continuous $MoS_2$ films were transferred from the catalyst surface using the above procedure onto 300 nm $SiO_2$/Si substrates with predefined electrical contacts[48]. Typical channel length was 100×100 μm². Devices were electrically characterised in a Linkam LTS600P probe station after desorbing water from the surface[49] via baking at 225 °C for 30 min in dry nitrogen. Subsequent measurements were performed under dry nitrogen at room temperature.

## Data Availability
The datasets generated during and/or analysed during the current study are available from the corresponding author on reasonable request.

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

## Acknowledgements

A.S., J.D.T., P.R.W, T.J.B. and P.B. acknowledge support from the EU Seventh Framework Programme (FP7/2007–2013) under grant agreement number FP7-6040007 GLADIATOR. D.M.A.M., P.R.W., and P.B. acknowledge support from the EU Horizon 2020 Future and Emerging Technologies programme under grant agreement numbers 696656 and 785219. A.S., J.D.T, M.G., N.S., T.J.B. and P.B. acknowledge support from the Danish National Research Foundation Center of Excellence for Nanostructured Graphene (CNG) (project DNRF103). R.-M.S., A.J.H, M.B. and P.H. acknowledge the support from the Danish Council for Independent Research, Natural Sciences under the Sapere Aude program (grant no. DFF-4002-00029) and Villum Fonden under Center of Excellence for Dirac Materials (grant no. 11744). A. Crovetto acknowledges support from the Villum Fonden, research grant 9455. M.G. and N.S. acknowledge support from the Danish National Research Foundation IDUN Center of Excellence (project DNRF122) and Villum Fonden (research grant 9301) for partially funding the PL setup.

## Author Contributions

A.S. conceived the method and performed the majority of the experiments. J.D.T. and T.J.B. performed TEM and SAED measurements. D.M.A.M. fabricated field-effect devices and performed electrical measurements. M.G. and N.S. performed and analysed photoluminescence measurements. R.-M.S., A.J.H, M.B., and P.H. performed LEED and ARPES characterisation. A. Crovetto did ellipsometry measurements. P.R.W. performed terahertz time domain spectroscopy measurements. A. Carvalho and A.H.C.N. provided band structure calculations. T.J.B. and P.B. supervised the project. T.J.B. and A.S. wrote the manuscript and prepared the figures, and all authors contributed to manuscript revisions.

## Additional information

**Competing interests:** A patent application has been filed and a PCT application has been published (WO 2018/087281 A1) by the Technical University of Denmark.

