## [Peer Review File · Nature Communications]

Reviewers' comments:

Reviewer #1 (Remarks to the Author):

The paper describes an interesting approach for the synthesis of binary 2D materials based on the formation of a Au-metal alloy film following by chemical conversion to a binary sulfide, selenide, telluride or nitride. A range of binary 2D materials are demonstrated, some of which are characterized using several techniques and others which are characterized only using XPS.

The synthesis method is relatively new for 2D materials but it is really a form of vapor-solid-solid (VSS) growth or perhaps even vapor-liquid-solid growth (VLS). In VLS, for example, a gold particle or film is initially heated with another material to form an alloy which, upon supersaturation, results in the precipitation of a solid phase. These methods have been used most recently for nanowire growth but the early work on VLS/VSS was focused on growing larger semiconductor crystals. Along similar lines to the current paper, binary materials such as GaAs, GaN, etc. are grown by VLS/VSS - for example, a Au-Ga alloy is formed and the As or N are not soluble in the gold alloy but incorporate on the surface to form the binary phase. The paper should reference these prior methods and describe the 2D materials growth within the context of a VSS or VLS-type of a process.

Along similar lines, the paper would be stronger if it included more of a mechanistic explanation for the growth. For example, an understanding of the Au-metal binary phase diagram is very important to predict which materials can be grown using this process. Looking at the Au-Mo phase diagram, for example, it is straightforward to see how this process could work, however, for Au-Nb, a liquid phase should form at the temperature employed but this isn't mentioned. Given the significant differences in the Au-metal phase diagrams, it is difficult to understand why the process should work using very similar conditions for all the different materials that were produced. The paper would be stronger if it contained an in-depth discussion of the processing fundamentals including phase diagrams for the respective Au-metal combinations.

Overall, this is an interesting approach for 2D materials synthesis, however, for publication in Nature Communications the authors should improve the scientific content of the paper and provide a more in-depth explanation of the growth mechanism based on the phase diagrams for the different Au-metal combinations.

Reviewer #2 (Remarks to the Author):

In this work, Abhay Shivayogimath and co-workers report a universal method for the synthesis of 2D binary compounds by using the alloying metal M films with gold. The various 2D binary compounds including S-, Se-, Te- and N-based materials are prepared. The structures and properties of certain kinds of materials have been characterized and studied. Although some binary materials are new, the method is not novelty and the size of as-produced samples is very small as well as the quality is poor. I recommend this manuscript to be reconsidered in Nature Communications after major revision. Please check the following comments, listed from most significant to least significant:

1. This method is not the first time to be reported on the synthesis of 2D TMDs using metal film alloys. Authors need to differentiate this work from the published papers like ReS₂/WS₂ heterostructures (Nature Communications 7: 13911 (2016)). The real novelty of this work should be illustrated more clearly and highlighted.
2. The sizes of most of the samples are relatively small, such as MoTe₂, VS₂, Cr-based, Co-based and Fe-based materials. It requires authors to confirm this method being suitable to synthesize large-size 2D TMDs for their applications. In addition, some samples are the first reported including Cr-based and Co-base binary compounds. The authors should discuss more about the Cr-based and Co-based binary materials rather than MoS₂ in the manuscript. Furthermore, the authors need to confirm the element ratio of Cr/Co and sulfur as 1:2 in the Cr-based and Co-based compounds using TEM as many possibilities of phases in Co- and Cr-based binary compounds such as CoS, CoS₂, Co₃S₄, and Co₈S₉.
3. The quality of the material is not very good and worse than what can be produced using other methods. The authors used the as-prepared MoS₂ to study the properties of monolayer MoS₂, however, the on/off in monolayer MoS₂ is very low. This indicates that the quality of as-synthesized MoS₂ is low. The authors need to explore the best recipe for obtaining the high quality of 2D binary. The low quality of the materials can also be seen from the AFM and SEM images of as-prepared materials, such as WS₂, WTe₂, MoSe₂, especially for the CoSe₂.

4. Many samples are very thick according to AFM images and SEM images. The authors should clearly show the data how many kinds of as-synthesized materials are monolayer and how the largest size of as-obtained samples.

5. The PL position in CVD grown WS₂ is less than 650 nm (Nano Lett., 2013, 13 (8), 3447). Why the position is around 680 nm? The authors have to confirm it. Meanwhile, why only one Raman peak exists in WTe₂?

6. The authors should clearly identify the phase of MoTe₂.

7. The authors mentioned that “salt additives have a detrimental impact on material performance through loss of epitaxy and alkali metal doping and substitution”. However, the description of doping and substitution is not correct (references see ACS Appl. Mater. Interfaces, 2018, 10 (47), 40831). Doping and substitution means the atoms incorporate into the lattice of the crystal. Absorption is not equal to the doping and substitution.

Reviewer #3 (Remarks to the Author):

The authors demonstrated a general approach for synthesizing thin layers of two-dimensional binary compounds, by depositing from elemental solid metal precursors. In this approach, a thin layer of metal M is sputtered onto a c-plane sapphire substrate and then sputter coated with a thick layer of gold. The Au-M alloy is subsequently exposed to a vapour-phase precursor of element X, resulting in few-atom thick layers of binary compounds. Using this method, the authors synthesized a wide range of 2D materials, including transition metal sulphides, selenides, tellurides and nitrides. Following are the comments and questions.

(1) There are many papers reported previously on synthesizing TMDs on Au, such as ACS Nano 2015, 9, 4, 4017-4025, Adv. Func. 2015, 25, 842-849, ACS Nano 2014, 8, 10, 10196-10204. The method proposed in this paper is closely related to or based on the previous method. It is important to cite those papers and identify the key differences, advantage and limitation of this proposed new method as compared to the previous method.

(2) Several TMDs, such as MoS₂, WS₂, WSe₂, have been successfully synthesized by CVD and MOCVD. In particular, researchers has demonstrated high-mobility 4-inch wafer-scale films of monolayer MoS₂ and WS₂ using MOCVD. It is not clear what is the advantage for this new method.

(3)The grain size of the TMDs synthesized using this method is very small. Although the crystal size is over 1 μm in Fig 2a, most crystal size is below 1 μm , as shown in Fig 2b, Fig S4-S9, Fig S12, and Fig S17-S18.

(4)In this method, TMDs are grown on gold. This implies that the TMD films need to be transferred out to an insulating substrate before the devices can be fabricated on these films. The transfer process can introduce resist/chemical residues, wrinkles and cracks of the film, which will degrade the performance, reduce yield, and increase the process cost. As a contrast, In CVD and MOCVD processes, TMDs can be directly grown on insulating substrate, which make it much more friendly for device fabrication.

(5)The authors mentioned that "The relative thicknesses of the M and Au layers determines the concentration of M in the final alloy, which here is deliberately limited to 5 at. % in order to maintain single-phase alloying conditions". Please provide the data to support this statement.

(6)There is only one IV curve shown in this paper. To confirm the electrical properties of these films, please provide the statistics of the carrier mobility for representative films, such as MoS_2 , WSe_2 .

(7)Please provide the AFM data of the synthesized film to show the thickness of the synthesized film.

(8)Please provide the Raman and PL mapping of the synthesized film to show the uniformity of the film.

(9)Please provide references for Line 54 ~ 56 on page 2.

Reviewers' comments:

Reviewer #1 (Remarks to the Author):

The paper describes an interesting approach for the synthesis of binary 2D materials based on the formation of a Au-metal alloy film following by chemical conversion to a binary sulfide, selenide, telluride or nitride. A range of binary 2D materials are demonstrated, some of which are characterized using several techniques and others which are characterized only using XPS.

We thank the reviewer for their close reading of our manuscript, and for the valuable feedback provided. We are glad that the reviewer judges our approach to be interesting.

The synthesis method is relatively new for 2D materials but it is really a form of vapor-solid-solid (VSS) growth or perhaps even vapor-liquid-solid growth (VLS). In VLS, for example, a gold particle or film is initially heated with another material to form an alloy which, upon supersaturation, results in the precipitation of a solid phase. These methods have been used most recently for nanowire growth but the early work on VLS/VSS was focused on growing larger semiconductor crystals. Along similar lines to the current paper, binary materials such as GaAs, GaN, etc. are grown by VLS/VSS - for example, a Au-Ga alloy is formed and the As or N are not soluble in the gold alloy but incorporate on the surface to form the binary phase. The paper should reference these prior methods and describe the 2D materials growth within the context of a VSS or VLS-type of a process.

The reviewer asks for the distinction between the presented growth technique and VLS/VSS type growths already reported in 1964 in 10.1063/1.1753975 and widely studied since then for the growth of 1D nanowires, 'whiskers' and other similar high aspect ratio nanostructures. The key differences we would highlight for the reviewer are as follows:

- VLS and VSS growth techniques rely on a supersaturation of one of the desired nanostructure components in the catalyst – e.g. Si in Au catalyst from a vapour phase SiCl_4 precursor, or Ga in Au catalyst in the case of GaAs or GaN nanowires. This supersaturation leads to precipitation of the solid nanostructure.
 - No such supersaturation is required here – indeed the process proceeds with only a few at. % of metal in the gold alloy and negligible solubility of the X component.
 - Saturation is actively avoided in order to allow surface limited growth of 2D materials.
- Wafer-scale full coverage 2D MX_n layers can here be grown in epitaxial alignment with the catalyst layer
 - We are not aware of any comparable VLS or VSS processes that can produce 100% coverage of an underlying substrate, due to the requirement of distinct catalyst particles or droplets dewetting both the substrate and the grown structure in VLS and VSS processes.
 - More generally, requirements and consideration of surface and interfacial energies and resulting contact angles between different phases are entirely eliminated in our growths.
- VLS or VSS growth is continuous as long as growth conditions are maintained.
 - Here, and similar to graphene chemical vapour deposition, the formation of a single layer is self-limiting and prevents further growth.

All of these points were addressed in the manuscript separately, but we have now also added an explanatory paragraph to the discussion on page 13:

“The method presented here is distinct from vapour-liquid-solid (VLS) and vapour-solid-solid (VSS) growth models³³ of III-V nanowires since we actively avoid supersaturation of component metals in gold – a key hallmark and necessary prerequisite for VLS/VSS growth mechanisms - by limiting the amount of metal available to <5 at%. This ensures that the resulting alloy behaves as a single-phase, simple solid solution at the growth temperature. The role of the gold here is to facilitate phase separation of M and X precursors such that they react only at the vapour-solid interface. Once this interface has been passivated by the 2D MX layer, the growth terminates, similar to the case of graphene on copper.”

Along similar lines, the paper would be stronger if it included more of a mechanistic explanation for the growth. For example, an understanding of the Au-metal binary phase diagram is very important to predict which materials can be grown using this process. Looking at the Au-Mo phase diagram, for example, it is straightforward to see how this process could work, however, for Au-Nb, a liquid phase should form at the temperature employed but this isn't mentioned. Given the significant differences in the Au-metal phase diagrams, it is difficult to understand why the process should work using very similar conditions for all the different materials that were produced. The paper would be stronger if it contained an in-depth discussion of the processing fundamentals including phase diagrams for the respective Au-metal combinations.

Our choice of 5 at.% was made to ensure a solid solution of M in Au and prevent the formation of multiple phases or intermetallic compounds, and was indeed informed by the phase diagrams of the Au-M systems – we agree we did not go into sufficient detail here in the manuscript and we are grateful to the reviewer for pointing to the need for a more careful explanation. The Au-Nb phase diagram presented in Okamoto, H. & Massalski, T.B. Bulletin of Alloy Phase Diagrams (1985) 6: 134 10.1007/BF02869225 is reproduced below:

Figure R1 – Taken from 10.1007/BF0286922. The star marks the growth temperature and 5 at.% alloy used.

At 5 at.% Nb in Au, at our growth temperature of 850°C and below the melting point, Nb is in solid solution with Au. This is the general case, and true for V (10.1007/10000866_331), Fe (10.1007/BF02868322), Co (10.1007/BF02869509), Cr (10.1007/10655491_69), Hf (10.1007/978-3-540-45280-5_29), Pt (10.1007/BF02871187), and Ta (10.1007/10000866_322).

Complete solid solutions are not formed in the cases of Mo, Re, and W. In these instances the gold layer dissolves some fraction of the metal. For Mo (10.1007/BF02867807) at 850°C a 1% Mo in Au phase is produced, plus a solid Mo phase with a negligible Au solubility at our growth temperature. Re (10.1007/BF02872959) is perhaps similar, with a 0.1% Re in Au phase suggested by resistivity measurements but no solubility suggested by other techniques. For W (10.1007/BF02869226) no known alloys are formed so W likely reaches the top surface only by diffusion.

Another subtlety suggested from the phase diagrams is the behavior on cooling in the case of incompletely reacted M components. Most solid solutions of 5 at.% remain so when cooling from 850°C (see Figure R1 here), but Fe and Co precipitate out on cooling (the solvus is crossed). This may explain the origin of some of the different structures we observe on the surface after growth. We have noted this behavior in the main text in the Discussion section:

“Controlling the M content can be particularly important in determining the layer coverage, maintaining single-phase alloying conditions, and in preventing precipitation of excess metal upon cooling. Such precipitation can interfere with 2D growth and lead to 3D structures on the surface, as seen for Co(21) or Fe(20).”

Overall, this is an interesting approach for 2D materials synthesis, however, for publication in Nature Communications the authors should improve the scientific content of the paper and provide a more in-

depth explanation of the growth mechanism based on the phase diagrams for the different Au-metal combinations.

We hope that the manuscript changes outlined above are satisfactory and provide more background detail on the mechanisms of the growth process, and thank the reviewer for their positive recommendation and helpful comments.

Reviewer #2 (Remarks to the Author):

In this work, Abhay Shivayogimath and co-workers report a universal method for the synthesis of 2D binary compounds by using the alloying metal M films with gold. The various 2D binary compounds including S-, Se-, Te- and N-based materials are prepared. The structures and properties of certain kinds of materials have been characterized and studied. Although some binary materials are new, the method is not novelty and the size of as-produced samples is very small as well as the quality is poor. I recommend this manuscript to be reconsidered in Nature Communications after major revision. Please check the following comments, listed from most significant to least significant:

1. This method is not the first time to be reported on the synthesis of 2D TMDs using metal film alloys. Authors need to differentiate this work from the published papers like ReS₂/WS₂ heterostructures (Nature Communications 7: 13911 (2016)). The real novelty of this work should be illustrated more clearly and highlighted.

We thank the reviewer for recommending consideration for publication after revision, and we hope the following comments address their concerns. We note that in response to the specific point raised by the reviewer, the suggested reference shown appeared as ref 19 in the submitted version, along with the following text highlighting the distinction between the methods and the novelty (emphasis added):

“We note that similar methods for growing atomically thin carbides and sulphides have been previously reported(18,19), but only for systems where both M and X components have limited solubility (< 0.1 at.%) in the catalyst. These represent a subset of the general approach presented here, i.e. one where the solubility of M in the catalyst approaches zero. In such cases, growth is dominated by surface-mediated diffusion of both M and X. This mechanism is distinct from that presented here, where instead the growth is dominated by precipitation of component M from the catalyst bulk upon reacting with X at the solid-gas interface at the surface of the catalyst. The crucial point is that surface-limited growth does not require the metal precursor to be insoluble in the catalyst layer. This fact allows our approach to be applicable to all precursor metals, in contrast to previous reports.”

2. The sizes of most of the samples are relatively small, such as MoTe₂, VS₂, Cr-based, Co-based and Fe-based materials. It requires authors to confirm this method being suitable to synthesize large-size 2D TMDs for their applications.

Our choice has been to study the MoS₂ produced by this scheme and present our results for this well-known material – optimization of the growth parameters for crystallite size for each demonstrated material in turn would take a prohibitive amount of time for one small team, and we also believe is beyond the scope of an initial report. Our results indicate that by extending the growth period it is possible to produce full coverage samples of all the presented materials. Control of nucleation is key, as is the case for graphene growth by CVD. The tendency to epitaxial alignment of the grown domains can help to mitigate these issues. We also note that for the purposes of this manuscript, full coverage layers

would provide no contrast variation in SEM images, step heights in AFM, etc. We will need more experimental work in the community – it has taken a decade and the work of many hundreds of researchers to develop CVD graphene science and technology to the level we see today, and making as many colleagues as possible aware of this new technique is the key reason we seek publication in Nature Communications.

In addition, some samples are the first reported including Cr-based and Co-base binary compounds. The authors should discuss more about the Cr-based and Co-based binary materials rather than MoS₂ in the manuscript.

Thank you for this suggestion. The novelty of these materials makes it very difficult to prove the efficacy of a new growth technique with them, and is the reason we choose to present results on MoS₂. We are glad that such a facile method is capable of growing these and other new materials, and we feel it is better to report the initial results, consisting of XPS, AFM and SEM and TEM data which confirm the structures rather than to omit these materials from the manuscript entirely, or make them the focus.

Furthermore, the authors need to confirm the element ratio of Cr/Co and sulfur as 1:2 in the Cr-based and Co-based compounds using TEM as many possibilities of phases in Co- and Cr-based binary compounds such as CoS, CoS₂, Co₃S₄, and Co₈S₉.

As described in the manuscript, transfer of some of the materials is quite challenging, with many of the materials being unstable in air or aqueous solutions, making the preparation of suspended TEM films very challenging. In the case of CoS₂, monolayer regions of the film are highly unstable under e-beam irradiation, as we have noted in the supplementary information. We therefore provide the presented stoichiometry for Cr/Co compounds based on the XPS data in the Supplementary Information.

3. The quality of the material is not very good and worse than what can be produced using other methods. The authors used the as-prepared MoS₂ to study the properties of monolayer MoS₂, however, the on/off in monolayer MoS₂ is very low. This indicates that the quality of as-synthesized MoS₂ is low. The authors need to explore the best recipe for obtaining the high quality of 2D binary. The low quality of the materials can also be seen from the AFM and SEM images of as-prepared materials, such as WS₂, WTe₂, MoSe₂, especially for the CoSe₂.

With regard to optimization of properties for each presented material, we would refer the reviewer to the response to point 2 above.

With specific reference to the on/off ratio for MoS₂, it is indeed somewhat low. This might be partly due to an effect of leakage through our SiO₂-based bottom gate insulator which prevents us from driving the devices to a saturated 'on' current. We have instead presented on off ratios for the highest achieved current in each case. Gate leakage most likely arises due to the need to transfer the MoS₂ devices by etching in a KI solution – K both dopes the MoS₂ and is a potential source of mobile ions in the gate oxide. The low on-off ratios could also potentially be due to residues from the transfer process. On the other hand we would highlight the following results as indicative of high quality:

- High mobilities 5-30 cm²/Vs
- Intensity of the photoluminescence peak being comparable to exfoliated MoS₂
- Lack of visible defects in TEM
- Long range order apparent in LEED

- ARPES
- Low FWHM of the Raman spectral peaks (corresponding to the wavenumber resolution measurement limit of our Raman system)

We believe on balance that the weight of evidence suggests that the presented route for synthesis is quite valuable, even without considering the ability to synthesize completely novel materials.

4. Many samples are very thick according to AFM images and SEM images. The authors should clearly show the data how many kinds of as-synthesized materials are monolayer and how the largest size of as-obtained samples.

With regard to optimization for size of domain or for monolayer coverage fraction, we would refer the reviewer to our response to point 2 above. Our response to reviewer 1 also describes the probable origin of the structures visible in the Co and Fe growths – namely a precipitation of excess metal upon cooling. Monolayer step heights measured by AFM on SiO₂ for WS₂, MoSe₂, WSe₂, and CoS₂ - which could be successfully transferred - are present in the Supplementary Information. Vanadium and tantalum nitrides were also successfully transferred, with associated AFM height profiles provided in the SI, but these are likely few nm thick crystalline layers rather than monolayers, as in the case of 2D Mo₂C as reported in DOI: 10.1038/nmat4374. We expect to be able to control layer thickness with further optimization in future work.

5. The PL position in CVD grown WS2 is less than 650 nm (Nano Lett., 2013, 13 (8), 3447). Why the position is around 680 nm? The authors have to confirm it.

The redshift is likely to be the result of doping induced by etching transfer, as is stated in the caption. We also note that the large asymmetry in the peak ratio of > 2 for the E_{12g}¹ and A_{1g} modes in the presented Raman spectrum for WS₂ also suggests a strong doping effect.

Meanwhile, why only one Raman peak exists in WTe2?

The peak position of the A_{1g} at 217 cm⁻¹ puts it at the edge of detectability in our Raman system, and is used as evidence of monolayer – the 217 cm⁻¹ is also known to be the strongest in WTe₂ [SI ref 14]. This peak shifts to smaller wavenumbers with increased layer numbers. Additional expected peaks around 160cm⁻¹ are therefore unresolvable vs noise. The low signal to noise ratio could also be a result of the small grain size here under the Raman spot, and the gold substrate can also have a strong influence on the Raman spectra obtained, with possibly a similar effect on the ratio of peaks as for WS₂.

6. The authors should clearly identify the phase of MoTe2.

None of the characterization techniques employed are sensitive to the different phases 1H/1T etc., and TEM which could be used to discriminate between these has not yet been possible for the reasons stated above in the answer to point 2. We agree that determining the phase of all of the presented materials would be highly desirable in the near future and would expect detailed characterization of each of the materials if the described technique is adopted by the community.

7. The authors mentioned that "salt additives have a detrimental impact on material performance through loss of epitaxy and alkali metal doping and substitution". However, the description of doping and substitution is not correct (references see ACS Appl. Mater. Interfaces, 2018, 10 (47), 40831). Doping

and substitution means the atoms incorporate into the lattice of the crystal. Absorption is not equal to the doping and substitution.

Thank you for helping us to correct this statement: indeed, the authors of that manuscript explicitly state they are unable to find evidence of substitutional doping. We have revised the sentence as follows:

“...which may have a detrimental impact on material performance through loss of epitaxy and alkali metal doping.”

Reviewer #3 (Remarks to the Author):

The authors demonstrated a general approach for synthesizing thin layers of two-dimensional binary compounds, by depositing from elemental solid metal precursors. In this approach, a thin layer of metal M is sputtered onto a c-plane sapphire substrate and then sputter coated with a thick layer of gold. The Au-M alloy is subsequently exposed to a vapour-phase precursor of element X, resulting in few-atom thick layers of binary compounds. Using this method, the authors synthesized a wide range of 2D materials, including transition metal sulphides, selenides, tellurides and nitrides. Following are the comments and questions.

We thank the reviewer for their kind words and close attention to our manuscript, along with their many helpful suggestions.

(1) There are many papers reported previously on synthesizing TMDs on Au, such as ACS Nano 2015, 9, 4, 4017-4025, Adv. Func. 2015, 25, 842-849, ACS Nano 2014, 8, 10, 10196-10204. The method proposed in this paper is closely related to or based on the previous method. It is important to cite those papers and identify the key differences, advantage and limitation of this proposed new method as compared to the previous method.

(2) Several TMDs, such as MoS₂, WS₂, WSe₂, have been successfully synthesized by CVD and MOCVD. In particular, researchers has demonstrated high-mobility 4-inch wafer-scale films of monolayer MoS₂ and WS₂ using MOCVD. It is not clear what is the advantage for this new method.

We thank the reviewer for these comments. The three reports listed in point (1) all employ the same growth technique which has been shown to produce excellent results for the growth of MoS₂ through the use of volatilized MoO_{3-x} suboxides and sulphur, with sulphurisation and growth of MoS₂ occurring on a downstream Au foil. However, the distinctions between that method and that presented here are quite strong – in particular the requirement of volatile M precursors. While demonstrated for Mo and W, we are not aware of any work showing that that scheme can be applied to many of the metal M components presented here. In our scheme, synthesis of a new MX_n material can be attempted by sputtering a controlled quantity of the elemental M prior to Au deposition, with the resulting quantity of MX_n produced on the target substrate thus predetermined. We also have no requirements for a carrier gas or concerns about gas phase reactions of precursor compounds as is typical for CVD processes, since the reaction can proceed using only elemental precursors – as such we have demonstrated the growth of samples using the same technique within UHV systems. The use of elemental precursors also eliminates any consideration of finding the correct M-containing volatile precursor gas in an MOCVD process, and allows a purer material to be formed. MOCVD processes can also require comparatively long growth times. In the manuscript we believe the reviewer may be referring to in point 2 - 10.1038/nature14417 - the stated growth time is 26 hours, as compared to < 20 minutes here.

We believe that the issue of TMDC growth is not yet solved to the degree that reporting alternative methods is not valuable, and that alternative synthesis techniques can still be considered, particularly where they are capable of yielding entirely new materials in a combinatorial fashion.

We have added references to the manuscript and comments to distinguish the work as follows:

“The use of gold as a catalyst for synthesizing 2D TMD layers has been previously reported in literature(12,13,34–36), however a key advantage of the present scheme is that solid elemental metal films are used as precursors rather than metal-organic vapours or volatile metal compounds, which makes our process significantly simpler to implement for both known and new TMDs.”

(3)The grain size of the TMDs synthesized using this method is very small. Although the crystal size is over 1 μm in Fig 2a, most crystal size is below 1 μm , as shown in Fig 2b, Fig S4-S9, Fig S12, and Fig S17-S18.

Many of the individual domains are on the order of 1 μm , however we expect these domains will stitch quite neatly due to epitaxial alignment with optimized growth times. Strictly speaking, Fig 2a shows a MoS_2 crystal of 6 μm side length, and Fig 5 shows a WS_2 crystal 25 μm side length, with the majority the materials presented in Fig 5 having side lengths larger than 1 μm . We would also note that some of the materials have been synthesized here for the first time. As described above, we believe optimization of growth parameters for each of the materials presented for maximum crystal size is beyond the scope of the present manuscript – we do look forward to working on this issue in subsequent reports.

(4)In this method, TMDs are grown on gold. This implies that the TMD films need to be transferred out to an insulating substrate before the devices can be fabricated on these films. The transfer process can introduce resist/chemical residues, wrinkles and cracks of the film, which will degrade the performance, reduce yield, and increase the process cost. As a contrast, In CVD and MOCVD processes, TMDs can be directly grown on insulating substrate, which make it much more friendly for device fabrication.

In the context of electrical FET-type devices and for growth on sapphire the reviewer is quite correct – however, we have demonstrated that transfer is indeed possible and the performance of the resulting devices is quite comparable to those obtained through other techniques aside from an unimpressive on/off ratio resulting from suboptimal transfer and excess doping. Nevertheless, transfer is not seen as an insurmountable problem in other systems, e.g. CVD growth of graphene on copper. Sapphire insulating layers – which would allow epitaxial growth of said materials - on Si back gate wafers are not yet commonly available, and direct growth on SiO_2 does not result in epitaxially aligned domains, requiring a salt assisted growth method for larger domain sizes, with the attendant potential device issues with gate leakage from salt ions (which can mean that synthesized layers must be transferred off of the growth oxide anyway).

We would also highlight the many interesting and technologically relevant applications of TMDCs on Au, in particular atomrystals, plasmonic applications, photovoltaics and catalytic applications, for which the present method would be particularly beneficial. A brief list of references where such systems are of interest is provided below:

10.1021/acs.nanolett.6b01914

10.1021/acsp Photonics.7b01103

10.1038/s41467-017-01398-3
10.1038/s41467-018-04934-x
10.1021/acs.nanolett.7b04342
10.1021/nn503211t
10.1021/acsnano.5b03199

(5)The authors mentioned that "The relative thicknesses of the M and Au layers determines the concentration of M in the final alloy, which here is deliberately limited to 5 at. % in order to maintain single-phase alloying conditions". Please provide the data to support this statement.

References to the literature for phase diagrams for each of the presented M-Au systems are now included in the manuscript.

(6)There is only one IV curve shown in this paper. To confirm the electrical properties of these films, please provide the statistics of the carrier mobility for representative films, such as MoS₂, WSe₂.

(7)Please provide the AFM data of the synthesized film to show the thickness of the synthesized film.

(8)Please provide the Raman and PL mapping of the synthesized film to show the uniformity of the film.

To confirm the electrical properties of MoS₂, the main subject of our studies, we have now provided a summary of the key properties of 10 devices total in the supplementary information including carrier mobility, doping level and on-off ratio. Please find AFM, Raman and PL mapping data of full coverage MoS₂ films below. We would highlight the uniformity of the A_{1g} and the E_{2g}¹ peak positions over the > 100x100 μm area shown, as well as the PL map which varies by little over 1% across the measured region. AFM also shows a layer height that corresponds well to a monolayer growth. We have omitted this information from the main manuscript as the maps are largely featureless but would be happy to include the data, perhaps in the Supplementary Information, if the reviewer judges them to be informative.

Figure R2 a,b) Raman spectral maps of MoS₂ film transferred to SiO₂. c) PL map of same. d) AFM height scan of scratched MoS₂ film on SiO₂. Inset: line profile of indicated region.

(9) Please provide references for Line 54 ~ 56 on page 2.

We have provided references as the reviewer recommends.

Reviewers' comments:

Reviewer #1 (Remarks to the Author):

The authors have satisfactorily addressed the comments concerning the mechanism of growth and comparison to VLS/VSS processes.

In their response, however, the authors state that this method enables wafer-scale full coverage of 2D MX_n layers but provide no evidence to support this. If the atomic % of the metal in gold is restricted to <5% and the Au film is thin, is there enough of the metal source available to fully cover the entire surface? The authors are claiming that this technique is superior to vapor phase synthesis methods such as MOCVD and MBE, consequently, they need to show evidence that it is indeed capable of achieving a fully coalesced epitaxial film over an area of at least 1x1 cm as has been demonstrated by these other methods.

Reviewer #2 (Remarks to the Author):

The author has almost addressed my concerns, but some comments are needed to be clarified using new data rather than only narrative answers, such as the low on/off in monolayer MoS₂ with a low quality. The authors said this is caused by the transferred method. However, there are many transferred methods can be used, for example, the method in Nature Communications, 2015, 6, 8569. The authors have to demonstrate the quality of the as-grown samples. Additionally, some comments in the revised version are subjective, for example, "this approach holds the advantages over state-of-the-art salt-assisted or metal-organic chemical vapor deposition (MOCVD) of 2D materials". More importantly, the growth in this manuscript is not strictly epitaxial synthesis as the grown samples are very small. The author should revise the manuscript carefully and address the concerns above.

Reviewer #3 (Remarks to the Author):

The authors have addressed all my questions. I would like to agree for publication.

Reviewer #1 (Remarks to the Author):

The authors have satisfactorily addressed the comments concerning the mechanism of growth and comparison to VLS/VSS processes.

We are glad that the reviewer is satisfied with our responses, and would like to thank them once again for their close reading and help in improving the manuscript.

In their response, however, the authors state that this method enables wafer-scale full coverage of 2D MXn layers but provide no evidence to support this. If the atomic % of the metal in gold is restricted to <5% and the Au film is thin, is there enough of the metal source available to fully cover the entire surface?

For an Au layer of 500 nm thickness - equivalent to $500 \text{ nm} / 0.22 \text{ nm} \sim 2000$ atomic Au layers - a single monolayer of M added to the surface would constitute only 0.05 at.%. By operating at 5 at.% we can safely assume – and indeed observe - that there is a sufficient quantity of M in the Au to allow for complete coverage without depleting the M from solid solution completely, independent of the lateral dimensions of the substrate.

The authors are claiming that this technique is superior to vapor phase synthesis methods such as MOCVD and MBE, consequently, they need to show evidence that it is indeed capable of achieving a fully coalesced epitaxial film over an area of at least 1x1 cm as has been demonstrated by these other methods.

As requested, as proof of the dimensions of the films, please find below optical images of a cm-scale continuous monolayer MoS₂ film transferred to oxide. Epitaxial alignment of the grown material to the Au before transfer is shown in figures in the manuscript 2b – SEM imaging of individual oriented domains prior to coalescence, 2c – LEED showing moiré pattern from MoS₂ on Au across the 1 mm spot size, and which we observe consistently across the sample, and 2d – bands visible in the ARPES data due to MoS₂ around the Au K point, again observable across the substrate surface, which would not be visible if the MoS₂ was not epitaxially aligned across the entire spot size. Indeed, both LEED and ARPES can easily reveal the presence of differently oriented domains within their respective spot size because these domains will lead to the simultaneous observation of several diffraction patterns / electronic dispersions, rotated against each other. Evidence for the high degree of crystalline order in the substrate is shown in supplementary Figure S1. It is also well known that fcc metals take on a {111} top surface on c-plane sapphire, e.g. 10.1016/S0039-6028(01)01685-5, 10.1021/acsnano.7b06196. MoS₂ is also known to grow epitaxially on Au facets, particularly the {111}, e.g. 10.1016/j.susc.2018.03.015, 10.1021/acs.langmuir.5b02533, 10.1039/C8NA00126J, 10.1021/acsnano.5b00081.

Figure 1: (a) photograph of a centimeter-scale continuous MoS₂ film transferred onto 300nm SiO₂/Si. (b) optical image of film in (a).

Reviewer #2 (Remarks to the Author):

The author has almost addressed my concerns, but some comments are needed to be clarified using new data rather than only narrative answers, such as the low on/off in monolayer MoS₂ with a low quality. The authors said this is caused by the transferred method. However, there are many transferred methods can be used, for example, the method in Nature Communications, 2015, 6, 8569.

We thank the reviewer for their suggestion for an alternate transfer method with proven excellent results for WS₂ on polycrystalline Au foils. We regret to say that we have already attempted this transfer method for our samples without success. This is of course disappointing, as such a transfer method would be ideal in our case, but as the authors of that study state, “the weak interaction between the monolayer WS₂ and the Au substrate for our samples enables the intact transfer of WS₂ from the substrate using the electrochemical bubbling method”. We believe that the strong binding between our MoS₂ films and the underlying Au(111) – possibly facilitated by our use of elemental precursors rather than metal-oxides, ultrasmooth surface of our Au(111) films, and strong Au-S bonding – is likely why this transfer process has so far failed in our case. Such transfer difficulties are known in literature for systems where the bonding between 2D material and its underlying substrate is strong, such as in that same paper in figures S14 and S15 for LPCVD WS₂ synthesis on Au, or for graphene on Ir (111) (10.1063/1.4958843).

We would very much like to obtain better FET measurements, and in particular better ON/OFF ratios for MoS₂ field effect devices, but we feel that optimising one figure of merit for one of the presented materials would be somewhat out of scope here given the wide range of applications beyond field effect devices. As described in our previous response the weight of evidence from SEM, TEM, LEED, ARPES, Raman, PL, XPS, FET mobility measurements, etc. also indicates that the as grown samples are of high quality.

The authors have to demonstrate the quality of the as-grown samples. Additionally, some comments in the revised version are subjective, for example, “this approach holds the advantages over state-of-the-art salt-assisted or metal-organic chemical vapor deposition (MOCVD) of 2D materials”.

We do indeed go on in that paragraph to list the specific advantages of our method over MOCVD, namely solid elemental precursors, lack of a need for salt additives, epitaxy and universality. Again, we believe that the experimental results provided – SEM, TEM, AFM, ARPES, LEED, PL, Raman, FET measurements etc. provide ample quantitative evidence of the quality of the grown materials.

More importantly, the growth in this manuscript is not strictly epitaxial synthesis as the grown samples are very small. The author should revise the manuscript carefully and address the concerns above.

We respectfully disagree that ‘epitaxial’ implies any specified areal coverage. To substantiate our use of the term in this context, we might point to published works such as e.g.

- 10.1021/nl1016706
- 10.1038/nmat4176
- 10.1038/nature01141
- 10.1038/nature01551
- 10.1021/nl047990x
- 10.1038/nmat2253

Any commonly accepted definition of 'epitaxy' will reduce to "the growth on a crystalline substrate of a crystalline substance that mimics the orientation of the substrate", and we would point to our response to reviewer 1 above where we show large-scale fully coalesced films transferred onto oxide and our SEM, ARPES and LEED results in support of this.

Reviewer #3 (Remarks to the Author):

The authors have addressed all my questions. I would like to agree for publication.

We thank the reviewer for their close attention and kind assistance in preparation of our manuscript for publication.